# CRISPR/Cas9-Mediated Knockout of *OsbZIP76* Reveals Its Role in ABA-Associated Immune Signaling in Rice

**DOI:** 10.3390/ijms26136374

**Published:** 2025-07-02

**Authors:** Yu-Jin Jung, Jin-Young Kim, Yong-Gu Cho, Kwon Kyoo Kang

**Affiliations:** 1Division of Horticultural Biotechnology, Hankyong National University, Anseong 17579, Republic of Korea; yuyu1216@hknu.ac.kr (Y.-J.J.); zino@hknu.ac.kr (J.-Y.K.); 2Institute of Genetic Engineering, Hankyong National University, Anseong 17579, Republic of Korea; 3Department of Crop Science, College of Agriculture and Life & Environment Sciences, Chungbuk National University, Cheongju 28644, Republic of Korea; ygcho@chungbuk.ac.kr

**Keywords:** *OsbZIP76*, CRISPR/Cas9, rice immunity, abscisic acid (ABA) signaling, pathogenesis-related genes (PR genes), *Xanthomonas oryzae* pv. *oryzae* (Xoo), *Magnaporthe oryzae*

## Abstract

The basic leucine zipper (bZIP) transcription factors are involved in a wide range of physiological processes in plants, including hormone signaling, stress responses, and growth and development regulation. They play a key role in abscisic acid (ABA)-mediated immune regulation. However, the immune-related function of *OsbZIP76* in rice remains poorly understood. In this study, we generated *OsbZIP76* knockout (KO) lines using CRISPR/Cas9-mediated genome editing and examined their phenotypic responses to the bacterial pathogen *Xanthomonas oryzae* pv. *oryzae* (Xoo) and the fungal pathogen *Magnaporthe oryzae*. The KO lines showed increased susceptibility to both pathogens compared to wild-type (WT) plants. Furthermore, qRT-PCR analysis revealed that, upon pathogen infection, the expression of pathogenesis-related genes such as *PR1a*, *PR5*, and *NPR1* was significantly suppressed in the KO lines. ABA treatment experiments showed that KO lines were hypersensitive to exogenous ABA, indicating a role for *OsbZIP76* in ABA perception and signaling. Notably, the expression of the *OsbZIP76* gene itself was strongly induced by both ABA treatment and pathogen infection, supporting its role as a positive regulator in ABA-associated immune signaling. Overall, this study demonstrates that *OsbZIP76* functions as an important immune regulator by integrating defense gene expression with ABA signaling, providing new insights into the molecular crosstalk between hormonal signaling and pathogen defense mechanisms.

## 1. Introduction

Plants in their natural environment are continuously exposed to a wide range of biotic stress factors such as pathogenic microorganisms, insects, and other herbivores, as well as various abiotic stresses including drought, salinity, extreme temperatures, oxidative stress, and nutrient deficiencies [1,2]. These stress factors disrupt physiological homeostasis, causing cellular damage, metabolic dysfunction, and growth inhibition, which ultimately reduce productivity and threaten plant survival [3,4]. Upon pathogen invasion, plants activate complex immune responses involving recognition of pathogenic effectors, signal transduction, and expression of defense-related genes to induce resistance [5]. Concurrently, abiotic stresses such as drought and salinity induce osmotic imbalance and oxidative damage at the cellular level, to which plants adaptively respond by regulating hormonal signaling pathways and antioxidant mechanisms [6,7]. Therefore, plants have evolved sophisticated physiological and molecular regulatory networks that integrate heterogeneous external stress signals, enabling active adaptation to environmental changes [1,6]. To effectively perceive and respond to diverse environmental stresses for survival and adaptation, plants have evolved intricate signaling networks that transmit external cues into intracellular responses. Among plant hormones, abscisic acid (ABA) serves as a key regulator primarily under abiotic stress conditions such as drought, high salinity, and low temperature [1,8]. ABA functions as a sensor for water deficit within the plant body, mediating physiological responses including stomatal closure, osmotic adjustment, and ionic homeostasis [8,9]. Furthermore, the ABA signaling pathway modulates the expression of stress-responsive genes to minimize cellular damage and enhance stress tolerance [10]. Beyond stress tolerance, ABA signaling also influences plant developmental processes and immune responses, contributing to optimized adaptation to fluctuating environments [1,11]. Recent studies have demonstrated that ABA, traditionally recognized for its role in abiotic stress tolerance, also functions as a pivotal modulator within plant immune systems [2,12]. ABA plays dual and context-dependent roles in defense regulation, modulating immune responses variably depending on pathogen type, infection site, and environmental context [13]. In some cases, ABA suppresses defense signaling pathways, increasing susceptibility to pathogens, whereas in others, it promotes immune activation, enhancing resistance [3,12]. This multifaceted regulation is attributed to ABA’s molecular interactions with salicylic acid (SA), jasmonic acid (JA), reactive oxygen species (ROS), and other hormonal and stress signaling pathways that fine-tune plant immunity [13,14]. The basic leucine zipper (bZIP) transcription factor family is one of the largest and most evolutionarily conserved groups in plants, acting as central regulators of diverse physiological and biochemical processes [15,16]. The bZIP proteins bind DNA and regulate transcription of target genes involved in seed development and maturation, signal transduction, stress responses, and hormone signaling—functions critical for plant survival and adaptation [5]. These factors form dimers through their leucine zipper domains, enabling specific binding to promoter regions to activate or repress transcription [15]. In rice (*Oryza sativa*), several bZIP transcription factors are closely associated with ABA signaling pathways and play crucial roles in regulating tolerance to drought and other abiotic stresses [9,17]. For example, *OsbZIP23* and *OsbZIP46* are activated by ABA and regulate ABA-responsive genes that mediate stomatal closure, osmotic adjustment, and antioxidant defense, thereby enhancing drought resistance in rice [9,10]. These transcription factors function as key components of the ABA signaling cascade, rapidly inducing gene expression changes in response to increased ABA levels under stress conditions [10]. Moreover, certain bZIP transcription factors also participate in biotic stress responses, mediating plant immunity against pathogens. *OsbZIP45* is a well-characterized example that contributes to resistance against rice blast disease caused by the fungal pathogen *Magnaporthe oryzae* (*M. oryzae*) [18,19]. The *OsbZIP45* mediates ABA- and ROS-dependent signaling pathways to regulate defense gene expression and amplify oxidative stress responses upon pathogen attack, leading to effective immune responses [19,20]. Thus, bZIP factors integrate hormonal and oxidative stress signals to coordinate environmental adaptation and immune defense [16,18]. Despite the growing understanding of bZIP factors as key regulators of stress signaling networks, the functional role of *OsbZIP76* remains largely uncharacterized. qRT-PCR data indicate that *OsbZIP76* expression is significantly induced by ABA and various abiotic stresses as well as hormone treatments. However, its specific molecular functions in pathogen resistance and immune regulation remain unclear. In particular, whether *OsbZIP76* serves as a molecular link integrating ABA signaling with immune response pathways is largely unknown. Detailed functional characterization of *OsbZIP76* will provide critical insights into the molecular crosstalk between plant hormone signaling and immune defense mechanisms.

In this study, we employed CRISPR/Cas9-mediated gene editing to systematically investigate the function of *OsbZIP76* in rice. Using pathogenic strains of *Xanthomonas oryzae* pv. *oryzae* (*Xoo*), which causes bacterial leaf blight, and *M. oryzae*, which causes rice blast disease, we assessed the immune phenotypes of knockout (KO) lines following pathogen infection. We also analyzed the expression of pathogenesis-related (PR) genes and the response of KO plants to exogenous ABA treatment to elucidate the regulatory role of *OsbZIP76* in hormone-mediated immune signaling. This research proposes *OsbZIP76* as a critical molecular regulator connecting ABA signaling and immune responses in rice, providing new insights into the complex interplay between plant hormones and pathogen defense signaling networks.

## 2. Results

### 2.1. Induction of OsbZIP76 Expression by ABA and Pathogen Infection

To investigate whether the expression of the *OsbZIP76* gene is induced by stress, rice seedlings were treated with 50 μM ABA, and the temporal expression levels of *OsbZIP76* were analyzed. Samples were collected at each time point, and qRT-PCR analysis revealed that *OsbZIP76* expression increased approximately 3.5-fold at 6 h after ABA treatment (Figure 1A). In addition, four-week-old rice seedlings were inoculated with *Xoo*, and gene expression was analyzed 12 h post-inoculation. The results showed that *OsbZIP76* expression was significantly higher in the *Xoo*-treated group compared to the mock control (Figure 1B).

### 2.2. Generation and Validation of OsbZIP76 Knockout Lines via CRISPR/Cas9

Given that *OsbZIP76* responded to both ABA treatment and *Xoo* infection in rice seedlings, we hypothesized that it may play a role in the response mechanisms to these stresses. To investigate the function of *OsbZIP76*, gene-edited rice plants were generated using the CRISPR/Cas9 system. The *OsbZIP76* gene (LOC_Os09g34880), comprising five exons and four introns, was targeted at the first and second exons by sgRNAs (Figure 2A). The constructed pBOsC vector harboring the sgRNAs was introduced into rice via *Agrobacterium tumefaciens* strain EHA105 (Appendix A). Transgenic plants were initially screened by PCR amplification of the nos/bar region (Appendix A), and successful gene-edited lines were identified through deep sequencing. Various mutation types, including homozygous, bi-allelic, and heterozygous edits, were observed (Figure 2B). Two representative lines were selected: bzip76 1-1 (homozygous T insertion) and bzip76 1-2 (bi-allelic A insertion/2 bp deletion). RT-qPCR analysis confirmed complete loss of *OsbZIP76* transcripts in these KO lines, indicating successful gene disruption (Figure 2C).

### 2.3. OsbZIP76 Knockout Results in Increased Susceptibility to Bacterial (Xoo) and Fungal Pathogens (M. oryzae)

To investigate the role of *OsbZIP76* in disease resistance, two independent KO lines (bzip76 1-1 and bzip76 1-2) and wild-type (WT) plants were inoculated with *Xoo* and *M. oryzae* on 4-week-old leaves. In response to *Xoo*, lesion lengths in bzip76 1-1 and bzip76 1-2 were 12.4 cm and 12.9 cm, respectively, whereas WT showed a lesion length of 6.3 cm (Figure 3A). Similarly, after *M. oryzae* infection, bzip76 1-1 and bzip76 1-2 exhibited more severe blast symptoms and higher disease severity scores compared to WT (Figure 3B). Quantitative analysis of bacterial and fungal biomass confirmed increased pathogen proliferation in both KO lines relative to WT (Figure 3C).

### 2.4. qRT-PCR Analysis of Pathogenesis-Related Gene Expression in bzip76 1-1 and bzip76 1-2 Lines

Based on the results from pathogen infection experiments with *Xoo* and *M. oryzae*, we analyzed the expression of key defense-related genes (*PR1a*, *PR5*, *NPR1*) to investigate the molecular basis of increased susceptibility in the KO lines (Figure 4). In WT plants, these genes were induced upon pathogen infection, whereas their induction was significantly reduced in the KO lines. Notably, *PR1a* expression decreased by more than 60% compared to WT (*p* < 0.01) (Figure 4). These findings suggest that disruption of *OsbZIP76* resulted in significantly reduced induction of defense-related genes (*PR1a*, *PR5*, *NPR1*) upon pathogen infection, indicating that *OsbZIP76* is required for a proper immune response.

### 2.5. Disruption of OsbZIP76 Affects ABA-Mediated Phenotypes

To investigate the involvement of *OsbZIP76* in ABA signaling, we examined physiological responses of KO lines and WT plants following exogenous ABA treatment. The KO lines (bzip76 1-1 and bzip76 1-2) exhibited markedly reduced leaf rolling, with scores decreasing from 4.5 in WT to 2.3 and 2.1, respectively (Figure 5A). Leaf water retention was also significantly lower in KO lines, dropping from 97.8% in WT to 58.7% and 61.2%, respectively, after 3 h of ABA treatment (Figure 5B). Furthermore, stomatal closure, a key ABA-regulated response, was impaired in KO lines, with average stomatal aperture widths increasing by ~31%, from 3.9 µm in WT to 5.1 µm and 5.0 µm, respectively (Figure 5C). These differences were statistically significant (*p* < 0.05). Collectively, these findings strongly suggest that *OsbZIP76* plays a crucial role in mediating ABA sensitivity, particularly by regulating stomatal behavior and dehydration tolerance mechanisms.

## 3. Discussion

The bZIP factors form one of the largest and most versatile TF families in plants, acting as key regulators that integrate hormonal and environmental signals to modulate gene expression. In rice, the subgroup of ABA-responsive bZIPs, including OsbZIP23, OsbZIP46, and OsbZIP72, has been intensively studied for their crucial roles in drought tolerance and ABA signaling pathways [19,21,22]. These TFs enhance plant survival under water deficit by activating downstream stress-responsive genes, promoting stomatal closure, and modulating ROS signaling. Despite their functional similarities, each bZIP TF may have unique or overlapping targets, contributing to a finely tuned stress response network [16,23,24,25]. Unlike well-characterized members, *OsbZIP76*’s biological function was not clearly defined until now. *OsbZIP76* encodes a bZIP-type transcription factor comprising three exons and a conserved DNA-binding domain, as revealed by genomic and domain analyses. Promoter motif scanning identified several cis-elements associated with stress and immune response, including ABRE and W-box motifs. Additionally, transcriptomic datasets indicate upregulation of *OsbZIP76* in response to pathogen infection, supporting its potential role in rice innate immunity. Our data reveal that *OsbZIP76* is strongly induced by both exogenous ABA treatment and pathogen infection by *Xoo* and *M. oryzae* (Figure 1 and Figure 2). This dual induction suggests a regulatory role bridging abiotic and biotic stress pathways, a feature shared by few bZIP family members. Such crosstalk is critical because plants must balance growth and defense, allocating resources depending on environmental cues. CRISPR/Cas9-mediated knockout of *OsbZIP76* significantly increased rice susceptibility to *Xoo* and *M. oryzae*, as demonstrated by longer lesion lengths and more severe blast symptoms compared to the WT (Figure 3). This phenotype strongly supports the positive role of *OsbZIP76* in activating immune defenses. Notably, the KO lines showed reduced expression of critical pathogenesis-related genes (*PR1a*, *PR5*, *NPR1*) after infection (Figure 4). *NPR1* is a master regulator of SA-mediated systemic acquired resistance (SAR) [26], and its downregulation in KO lines implicates *OsbZIP76* in modulating SA-dependent pathways. The interplay between ABA and other phytohormones is complex, exhibiting both antagonistic and synergistic interactions depending on the spatial and temporal context [18,27,28,29]. Our finding that *OsbZIP76* affects both ABA signaling and SA-regulated immune responses suggests that it might function as a molecular node coordinating these pathways. This coordination is likely crucial for optimizing defense without compromising growth under fluctuating environmental conditions. The repression of PR genes in KO plants, despite pathogen challenge, underscores the importance of *OsbZIP76* in sustaining an effective immune response. In addition to immune phenotypes, *OsbZIP76* KO lines exhibited altered physiological responses to ABA. The KO lines had reduced leaf rolling and water retention under ABA treatment, indicating impaired drought response (Figure 5A,B). Furthermore, stomatal assays showed that KO plants had significantly larger stomatal apertures upon ABA treatment, reflecting diminished stomatal closure capacity (Figure 5C). Stomatal closure serves as a primary defense mechanism to restrict pathogen entry and conserve water during drought stress. Recent studies have highlighted the critical role of ABA in mediating stomatal closure, emphasizing its importance in plant immunity and drought tolerance [16,23,30,31]. These findings underscore the necessity of *OsbZIP76* for proper ABA sensitivity and stomatal regulation, which are essential for effective defense responses. This may explain the enhanced susceptibility of KO lines to pathogens. The mechanism by which *OsbZIP76* modulates these processes remains to be fully elucidated. The bZIP proteins typically bind to ABA-responsive elements (ABREs) in target gene promoters, either as homodimers or heterodimers with other bZIP or non-bZIP TFs [4,7,16,24]. Identifying *OsbZIP76* direct target genes via chromatin immunoprecipitation sequencing (ChIP-seq) would clarify its transcriptional network. Moreover, potential interactions with other stress-related TF families such as NAC, WRKY, or TGA could broaden its regulatory influence. Given the dual role in abiotic and biotic stress, *OsbZIP76* may also influence ROS signaling and callose deposition, both crucial for pathogen defense and ABA responses [15,16,30,32,33]. Also, *OsbZIP76* may influence ROS production during pathogen infection. As several bZIP transcription factors are known to regulate ROS-related defense responses, *OsbZIP76* could contribute to oxidative burst and signaling. Integrating transcriptomic and proteomic data under combined drought and pathogen stress could reveal additional layers of regulation involving *OsbZIP76*. The dual-function nature of *OsbZIP76* makes it a promising target for engineering rice varieties with improved tolerance to multiple stresses. By modulating *OsbZIP76* activity, it may be possible to enhance resistance to devastating pathogens while maintaining or improving drought tolerance. This integrated stress resilience is vital under current climate change scenarios, where plants face simultaneous abiotic and biotic challenges.

## 4. Materials and Methods

### 4.1. Plant Materials and Growth Conditions

Rice (*Oryza sativa* L. ssp. *japonica* cv. Dongjin) was used as the genetic background for all experiments, including gene editing, phenotypic analysis, pathogen infection, and hormone response assays. Seeds were surface-sterilized with 70% ethanol for 1 min followed by 2.5% sodium hypochlorite for 15 min and then rinsed thoroughly with sterile distilled water. Sterilized seeds were germinated on half-strength Murashige and Skoog (½ MS) agar medium for 5 days under controlled conditions. Germinated seedlings were transplanted into soil-filled pots (1:1:1 mixture of peat moss, vermiculite, and perlite) and grown in a walk-in growth chamber with precisely controlled environmental settings: 16 h light/8 h dark photoperiod, constant temperature of 28 ± 1 °C, and relative humidity maintained at 60–70%. Light intensity was approximately 200 μmol m^−2^ s^−1^ provided by fluorescent lamps. For pathogen infection assays (with *Xoo* and *M. oryzae*) and exogenous hormone treatments (ABA), three- to four-week-old seedlings at the 3–4 leaf stage were selected to ensure consistent developmental stage and physiological status. Plants were randomly assigned to treatment and control groups to minimize positional and environmental bias. All experiments were repeated at least three times using independently grown biological replicates.

### 4.2. Gene Editing by CRISPR/Cas9 System

The full genomic sequence of *OsbZIP76* was retrieved from the NCBI database (https://blast.ncbi.nlm.nih.gov; accessed on 12 August 2024). Guide RNA (gRNA) target sites were designed within exon regions of *OsbZIP76*, ensuring proximity to the protospacer adjacent motif (PAM) sequence, using the CRISPR RGEN Tools (http://www.rgenome.net/; accessed on 2 September 2024) developed by Hanyang University [34]. Two specific single guide RNAs (sgRNAs) with minimal predicted off-target effects were selected (Appendix A). Custom oligonucleotide pairs for each sgRNA were synthesized (Bioneer Co., Ltd., Daejeon, Republic of Korea), annealed, and cloned into the binary vector pBOsC containing a Cas9 expression cassette via *AarI* restriction enzyme sites. The resulting constructs were introduced into *Agrobacterium tumefaciens* strain EHA105 and subsequently used to transform rice (*Oryza sativa* ssp. *japonica* cv. Dongjin) embryogenic calli by the *Agrobacterium*-mediated transformation method as previously described [35]. Following infection and co-cultivation, the calli were cultured and subcultured on 2N6 medium for selection and proliferation [36]. Transgenic calli were screened on selection medium containing 6 mg/L phosphinothricin (PPT) and 400 mg/L carbenicillin to ensure stable T-DNA integration and bacterial elimination. Regenerated shoots were rooted and acclimatized, then transferred to soil-filled pots and grown under greenhouse conditions for further analysis.

### 4.3. Detection of Mutation Type

To confirm CRISPR/Cas9-induced mutations in *OsbZIP76*, genomic DNA was extracted from approximately 100 mg of fresh rice leaf tissue using the DNA Quick Plant Kit (Inclone, Jeonju, Republic of Korea), following the manufacturer’s protocol. To identify T-DNA insertion lines, PCR screening was conducted using primers specific to the nos/bar selectable marker gene, as described previously [37]. PCR-positive lines were considered putative transgenic plants. To determine the nature and efficiency of gene editing events at the targeted *OsbZIP76* loci, site-specific regions flanking each sgRNA target site were PCR-amplified from genomic DNA and subjected to next-generation sequencing (NGS) using the Illumina MiniSeq platform (San Diego, CA, USA) with paired-end reads. Raw sequencing data were processed and analyzed using Cas-Analyzer (http://www.rgenome.net/cas-analyzer/#!; accessed on 8 December 2024), an online tool for evaluating CRISPR-induced mutations [38,39]. The mutation spectrum, such as insertions, deletions, and substitutions, was characterized for each line, and homozygous, heterozygous, and biallelic mutant genotypes were identified accordingly. The primers used in the experiment are described in Appendix A.

### 4.4. Pathogen Inoculation Assays

#### 4.4.1. *Xanthomonas oryzae* pv. *oryzae*

To evaluate bacterial blight resistance, the virulent Korean strain *Xoo* KACC10331 was used. The bacterium was streaked on peptone sucrose agar (PSA; 10 g/L peptone, 10 g/L sucrose, 1 g/L glutamic acid, 15 g/L agar) and incubated at 28 °C for 3 days. A single bacterial colony was inoculated into peptone sucrose broth and cultured overnight with shaking at 28 °C until it reached an OD_600_ of 0.5 (~10^8^ CFU/mL). Fully expanded leaves of 4-week-old rice plants were inoculated using the standard leaf-clipping method. Approximately the top 2–3 cm of each leaf was clipped with scissors dipped in the bacterial suspension. After inoculation, plants were maintained under standard growth conditions (28 ± 1 °C, 70% relative humidity, 16 h light/8 h dark). Disease severity was assessed at 14 days post-inoculation (dpi) by measuring lesion lengths from the cutting site downward. To quantify bacterial populations, infected leaf segments (1 cm) were excised, surface-sterilized, ground in sterile water, and serially diluted before plating on PSA. Colonies were counted after 3 days of incubation at 28 °C.

#### 4.4.2. *Magnaporthe oryzae*

To assess resistance to rice blast, the *M. oryzae* isolate KJ201 was used. The fungus was grown on oatmeal agar (OMA; 30 g/L oatmeal, 15 g/L agar) at 25 °C under constant fluorescent light for 10 days to induce sporulation. Spores were harvested by flooding the plates with sterile distilled water containing 0.02% (*v*/*v*) Tween 20 and gently scraping the colony surface. The resulting spore suspension was filtered through two layers of Miracloth and adjusted to a final concentration of 5 × 10^5^ conidia/mL. Three-week-old rice seedlings were uniformly sprayed with the spore suspension using an airbrush until leaf surfaces were fully covered. Immediately after inoculation, plants were enclosed in a transparent plastic box to maintain high humidity and incubated in darkness at 25 °C for 24 h to promote infection. After incubation, plants were returned to a standard growth chamber (16 h light/8 h dark, 28 °C, 70% humidity). Disease symptoms were evaluated at 7 dpi by scoring lesion number and type (spindle-shaped, water-soaked necrotic spots) on the second and third leaves.

### 4.5. ABA Treatment and Stomatal Response Assay

To evaluate the physiological response to ABA, 3-week-old rice plants were used. Fully expanded leaves from each genotype were excised at the base and immediately immersed in petri dishes containing either 50 µM (±)-ABA (Sigma-Aldrich, St. Louis, MO, USA) prepared in MES buffer (10 mM MES, pH 6.15, with 50 mM KCl) or the same buffer containing 0.1% ethanol as a mock control. Detached leaves were incubated under white light (~120 μmol photons m^−2^ s^−1^) at 25 °C for 6 h to allow for ABA-induced responses.

#### 4.5.1. Leaf Rolling and Water Loss Assay

Leaf rolling was visually scored at the end of the incubation period based on degree of curvature and rigidity, and representative phenotypes were photographed. For leaf water loss analysis, detached leaves were weighed immediately after excision (initial fresh weight) and again after 6 h of ABA or mock treatment. Relative water loss was calculated as a percentage of the initial fresh weight.

#### 4.5.2. Stomatal Aperture Measurement

To observe ABA-induced stomatal closure, leaf abaxial epidermal strips were manually peeled from the treated leaves using forceps. The epidermal strips were immediately fixed in 3:1 ethanol–acetic acid solution for 30 min, rehydrated with distilled water, and mounted on microscope slides. Stomatal apertures (width and length) were imaged using a light microscope (e.g., Olympus BX53, Tokyo, Japan) equipped with a calibrated micrometer. For each treatment and genotype, at least 100 stomata from 3 biological replicates were measured using ImageJ software (version 1.53t, NIH, Bethesda, MD, USA). Stomatal aperture was expressed as the ratio of width to length (W/L), and statistical analysis was performed using Student’s *t*-test or ANOVA, as appropriate.

### 4.6. RNA Extraction and Quantitative Real-Time PCR (qRT-PCR)

For gene expression analysis, total RNA was extracted from approximately 100 mg of fully expanded rice leaf tissue using the RNeasy Plant Mini Kit (Qiagen, Hilden, Germany) according to the manufacturer’s instructions. RNA integrity and concentration were assessed using agarose gel electrophoresis and a NanoDrop 2000 spectrophotometer (Thermo Fisher Scientific, Waltham, MA, USA), respectively. To eliminate potential genomic DNA contamination, 1 µg of total RNA was treated with RNase-free DNase I (Qiagen) at 37 °C for 30 min. The DNase-treated RNA was then reverse-transcribed into cDNA using the SuperScript III First-Strand Synthesis System (Invitrogen, Carlsbad, CA, USA) with oligo(dT) primers in a total reaction volume of 20 µL, following the manufacturer’s protocol. qRT-PCR was carried out using a QuantStudio 6 Flex Real-Time PCR System (Applied Biosystems, Foster City, CA, USA) and TB Green Premix Ex Taq II (Tli RNaseH Plus) (Takara Bio, Shiga, Japan). Each 20 μL reaction mixture included 1 μL of diluted cDNA template (1:10), 0.4 μM each of forward and reverse primers, and 10 μL of TB Green master mix. The thermal cycling conditions were as follows: initial denaturation at 95 °C for 30 s, followed by 40 cycles of 95 °C for 5 s and 60 °C for 30 s. A melting curve analysis was performed at the end of each run to verify amplification specificity. Expression levels of target genes, including *PR1a*, *PR5*, *NPR1*, and *OsbZIP76*, were normalized against the internal reference gene *OsUBQ5* (Ubiquitin 5), which was confirmed to be stable under our experimental conditions. All reactions were conducted in three biological and three technical replicates. The relative expression levels were calculated using the 2^−ΔΔCt^ method. Primer sequences used for qRT-PCR are provided in Appendix A.

### 4.7. Statistical Analysis

All experimental data were obtained from at least three independent biological replicates, with each biological replicate comprising multiple technical replicates where applicable. Quantitative results, including gene expression levels, lesion lengths, bacterial populations, stomatal aperture measurements, and water loss rates, were expressed as means ± SD. For statistical evaluation, one-way analysis of variance (ANOVA) was performed to test for significant differences among multiple groups. Where the ANOVA indicated significance, Tukey’s Honest Significant Difference (HSD) post hoc test was applied for pairwise comparisons. A *p*-value < 0.05 was considered statistically significant. All statistical analyses and graph generation were conducted using GraphPad Prism version 9.0 (GraphPad Software, San Diego, CA, USA). The assumptions of normality and homogeneity of variance were checked prior to applying parametric tests.

## 5. Conclusions

In summary, our findings establish *OsbZIP76* as a central regulator of ABA-associated immune responses in rice. CRISPR/Cas9-mediated knockout of *OsbZIP76* led to a lesion length increase from 6.3 cm in WT to 12.4~12.9 cm in KO lines, representing a 2-fold enhancement in disease susceptibility to Xoo. Likewise, blast disease symptoms caused by *M. oryzae* were markedly more severe in KO lines. The expression of key defense-related genes (*PR1a*, *PR5*, and *NPR1*) was reduced by 61.8–68.3% in KO lines relative to WT upon pathogen infection, indicating that *OsbZIP76* positively regulates PR gene induction during immune activation. Physiological assays further revealed that *OsbZIP76* knockout impairs ABA sensitivity. KO plants exhibited ~40% lower leaf water retention (58.7% in KO vs. 97.8% in WT) and less pronounced leaf rolling after ABA treatment. Moreover, average stomatal aperture widths were 5.1 µm in KO lines vs. 3.9 µm in WT, indicating a ~30% reduction in ABA-induced stomatal closure. Consistent with these phenotypes, *OsbZIP76* transcript levels were induced up to 3.5-fold within 6 h of exogenous ABA treatment and also upregulated after pathogen exposure. Collectively, our data reveal that *OsbZIP76* integrates abiotic and biotic stress signaling by modulating both ABA sensitivity and immune gene expression. These findings highlight *OsbZIP76* as a promising target for the development of rice cultivars with enhanced resistance to both pathogen infection and drought-related stress, which is increasingly relevant under climate change conditions.

## Figures and Tables

**Figure 1 ijms-26-06374-f001:**
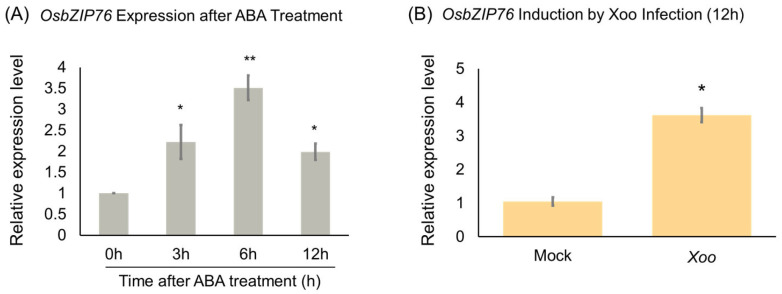
Expression of *OsbZIP76* in response to ABA treatment and Xoo infection in rice seedlings. (**A**) qRT-PCR analysis showing temporal induction of *OsbZIP76* expression following 50 μM ABA treatment. (**B**) *OsbZIP76* expression levels 12 h after inoculation with *Xoo*. Expression values were normalized to a housekeeping gene and are presented relative to untreated or mock-treated controls. Bars represent mean ± standard deviation (SD) from three independent biological replicates (*n* = 3). (*) Asterisks indicate statistically significant differences compared to controls (* *p* < 0.05, ** *p* < 0.01).

**Figure 2 ijms-26-06374-f002:**
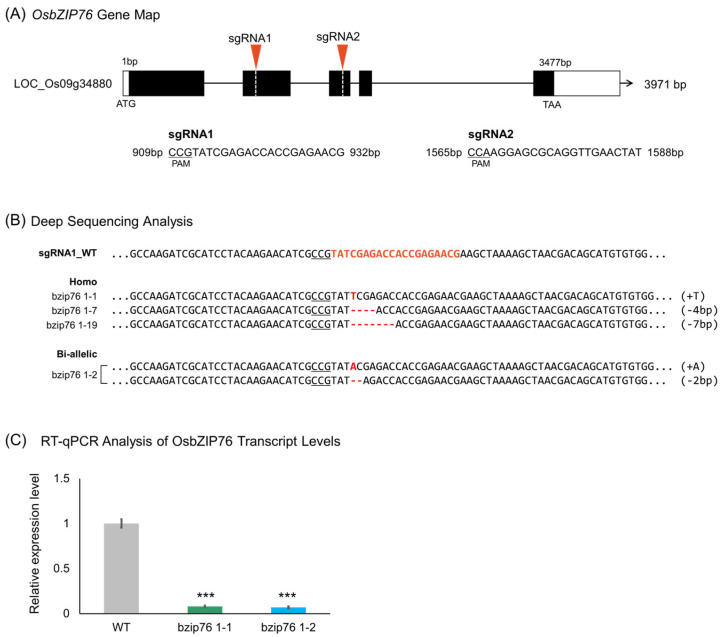
Generation and molecular validation of *OsbZIP76* KO lines using CRISPR/Cas9 in rice. (**A**) Schematic diagram of the *OsbZIP76* gene structure showing the positions of two sgRNA target sites used for CRISPR/Cas9 editing. Red triangles indicate Cas9 cleavage sites. (**B**) Representative mutation patterns identified by targeted deep sequencing. Orange text indicates the sgRNA sequences, while red letters represent nucleotide insertions or deletions (indels) introduced by CRISPR/Cas9 editing. (**C**) RT-qPCR analysis confirms the loss of *OsbZIP76* transcript in KO lines. Bars represent mean ± SD from three independent biological replicates (*n* = 3). (*) Asterisks indicate statistically significant differences compared to WT (*** *p* < 0.001).

**Figure 3 ijms-26-06374-f003:**
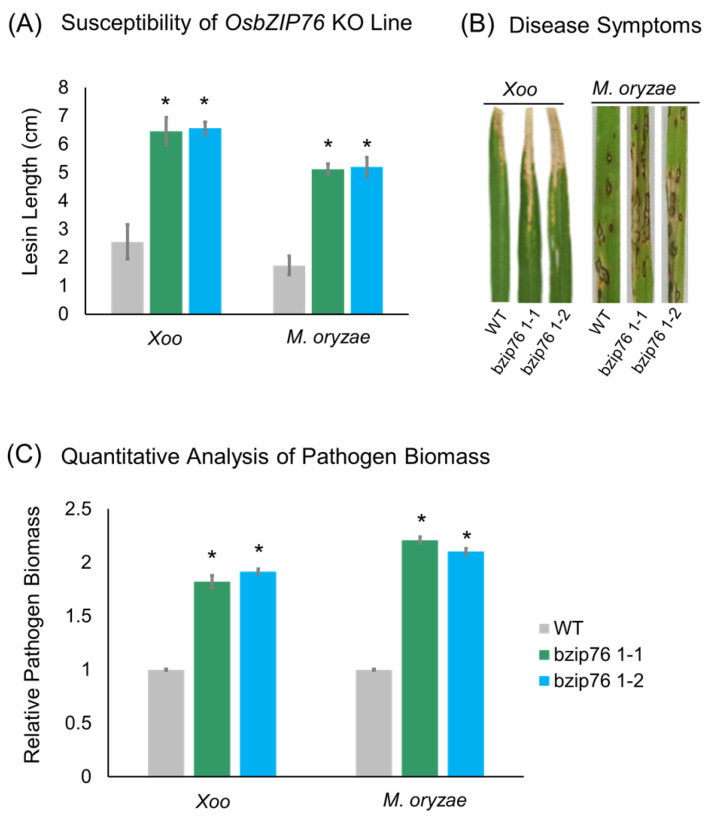
The *OsbZIP76* KO lines exhibit increased susceptibility to *Xoo* and *M. oryzae* infection (**A**) Lesion length measurements on 4-week-old rice leaves inoculated with *Xoo* show significantly longer lesions in *OsbZIP76* knockout (bzip76 1-1 and bzip76 1-2) lines compared to WT. Data represent mean ± SD from at least 10 leaves per line. (**B**) Disease symptoms and severity scores after *M. oryzae* infection reveal more severe blast symptoms and higher disease scores in KO lines relative to WT. (**C**) Quantification of bacterial and fungal biomass demonstrates increased pathogen proliferation in KO lines compared to WT plants. Bars represent mean ± SD from three independent biological replicates (*n* = 3). (*) Asterisks indicate statistically significant differences compared to WT (* *p* < 0.05).

**Figure 4 ijms-26-06374-f004:**
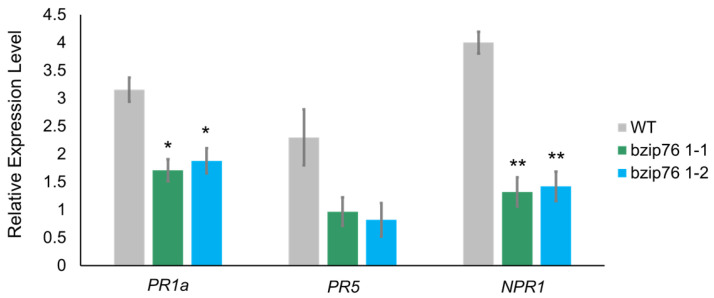
Expression analysis of defense-related genes in *OsbZIP76* KO and WT plants after pathogen infection. Bars represent mean ± SD from three independent biological replicates (*n* = 3). (*) Asterisks indicate statistically significant differences compared to WT (* *p* < 0.05, ** *p* < 0.01).

**Figure 5 ijms-26-06374-f005:**
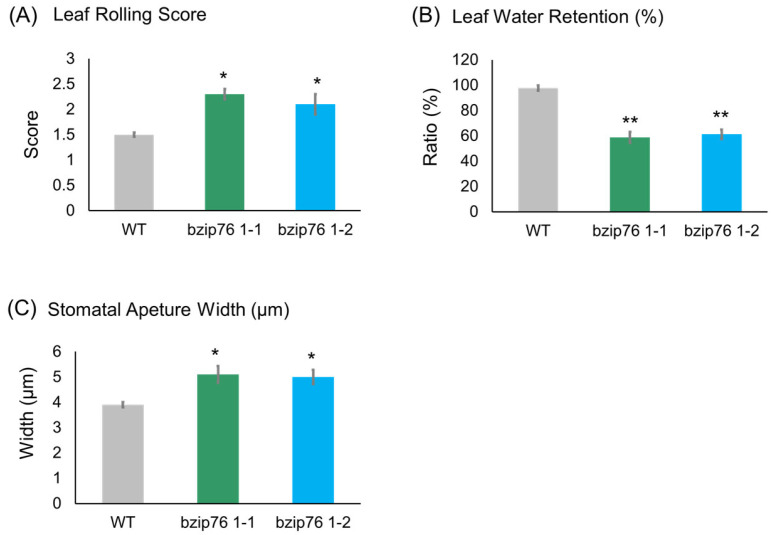
The *OsbZIP76* knockout lines exhibit reduced sensitivity to ABA. (**A**) Leaf rolling scores after 3 h treatment with 50 μM ABA. KO lines (bzip76 1-1 and bzip76 1-2) showed significantly reduced rolling (4.5 in WT vs. 2.3 and 2.1 in KO). (**B**) Leaf water retention (%) following ABA treatment. WT plants retained 97.8% of their initial weight, while KO lines retained 58.7% and 61.2%, respectively. (**C**) Stomatal aperture widths measured after ABA exposure. KO lines exhibited impaired closure, with apertures increasing from 3.9 µm in WT to 5.1 µm and 5.0 µm (~31% increase). Bars represent mean ± SD from three independent biological replicates (*n* = 3). (*) Asterisks indicate statistically significant differences compared to WT (* *p* < 0.05, ** *p* < 0.01).

## Data Availability

The original contributions presented in this study are included in the article/Appendix A; further inquiries can be directed to the corresponding authors.

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
