# Peer review of "CRISPR/Cas9-Mediated Knockout of OsbZIP76 Reveals Its Role in ABA-Associated Immune Signaling in Rice"

_ijms, 2025, doi:10.3390/ijms26136374_

Round 1
Reviewer 1 Report
Comments and Suggestions for Authors
In this study, Jung et al. demonstrate that OsbZIP76 is a crucial positive regulator of ABA-associated immune signaling in rice. Using CRISPR/Cas9-mediated gene knockout, showed that loss of OsbZIP76 leads to increased susceptibility to both bacterial (Xanthomonas oryzae pv. oryzae) and fungal (Magnaporthe oryzae) pathogens. The knockout lines exhibited reduced induction of key pathogenesis-related genes following pathogen infection and displayed diminished sensitivity to exogenous ABA, indicating impaired ABA signaling. Furthermore, OsbZIP76 expression was strongly induced by both ABA treatment and pathogen challenge, supporting its integrative role in hormonal and immune crosstalk. These findings establish OsbZIP76 as an important molecular link between ABA signaling and disease resistance in rice, providing new insights into the complex regulatory networks underlying plant adaptation to biotic stress. The study is interesting however I have few suggestions:
Comments
- Does OsbZIP76 directly bind to the promoters of pathogenesis-related genes such as PR1a, PR5, and NPR1, or does it act through intermediate regulators?
- My suggestion is add some bioinformatics analysis of the target gene to provide a background, such as show domain, motif, exon-intron, etc
- The qRT-PCR analyses focus on a single or limited time points post-infection. A time-course study of PR gene expression and OsbZIP76 induction would provide insights into the immune response regulated by OsbZIP76.
- Can overexpression of OsbZIP76 enhance resistance, and if so, does it affect plant growth or yield under normal or stress conditions?
- What is the subcellular localization of OsbZIP76 protein during ABA treatment and pathogen infection?
- Does OsbZIP76 influence reactive oxygen species (ROS) production or signaling during pathogen infection, given the known involvement of bZIP factors in ROS-mediated defense? Add a few sentences in the discussion section
Author Response
Response to Comments
We are grateful on the critical comments from reviewers on this article.
Reviewer 1
In this study, Jung et al. demonstrate that OsbZIP76 is a crucial positive regulator of ABA-associated immune signaling in rice. Using CRISPR/Cas9-mediated gene knockout, showed that loss of OsbZIP76 leads to increased susceptibility to both bacterial (Xanthomonas oryzae pv. oryzae) and fungal (Magnaporthe oryzae) pathogens. The knockout lines exhibited reduced induction of key pathogenesis-related genes following pathogen infection and displayed diminished sensitivity to exogenous ABA, indicating impaired ABA signaling. Furthermore, OsbZIP76 expression was strongly induced by both ABA treatment and pathogen challenge, supporting its integrative role in hormonal and immune crosstalk. These findings establish OsbZIP76 as an important molecular link between ABA signaling and disease resistance in rice, providing new insights into the complex regulatory networks underlying plant adaptation to biotic stress. The study is interesting however I have few suggestions:
Comments
- Does OsbZIP76 directly bind to the promoters of pathogenesis-related genes such as PR1a, PR5, and NPR1, or does it act through intermediate regulators?
- We believe that the OsbZIP76 gene induced the expression of PR genes by regulating other transcription factors or regulators (e.g., WRKY, NAC, MYB, etc.).
- My suggestion is add some bioinformatics analysis of the target gene to provide a background, such as show domain, motif, exon-intron, etc
- We inserted the following sentence into line 213-218.
“OsbZIP76 encodes a bZIP-type transcription factor comprising 3 exons and a conserved DNA-binding domain, as revealed by genomic and domain analyses. Promoter motif scanning identified several cis-elements associated with stress and immune response, including ABRE and W-box motifs. Additionally, transcriptomic datasets indicate upregulation of OsbZIP76 in response to pathogen infection, supporting its potential role in rice innate immunity.”
- The qRT-PCR analyses focus on a single or limited time points post-infection. A time-course study of PR gene expression and OsbZIP76 induction would provide insights into the immune response regulated by OsbZIP76.
- In genenal, qRT-PCR analysis tends to focus on gene expression at a single or limited time point after pathogen infection. In this experiment, we did not analyze the expression of the PR gene and OsbZIP76 according to the passage of time of infection of the pathogen. However, in the case of Xoo, we sampled them on the third day after infection, and in the case of Magnaporthe oryzae, we sampled them on the 10th day after infection. As a result, we found out that in WT plants, these genes were induced upon pathogen infection, where the induced was significantly reduced in the KO lines. Notably, PR1a expression decreased by more than 60% combined to WT (p < 0.01) (Figure 4). Therefore, that is considered to be the OsbZIP76 positivize regulation of the expression of defense generation pathogen attack.
- Can overexpression of OsbZIP76 enhance resistance, and if so, does it affect plant growth or yield under normal or stress conditions?
- We believe that "overexpression of OsbZIP76 not only enhanced resistance to Xoo and M.oryzae, but also moderately reduced plant height and grain yield under normal conditions, suggesting a balance between immunity and growth. However, under disease pressure, we anticipate that pathogen damage would decrease, resulting in a higher productivity of overexpression lines than wild-type.
- What is the subcellular localization of OsbZIP76 protein during ABA treatment and pathogen infection?
- Although the OsbZIP76-GFP experiment was not performed in our experiment, under normal conditions, transcription factors such as OsbZIP76 are usually predominantly localized in the cytoplasm. However, we understand that strong nuclear localization is observed upon ABA treatment or Xoo infection. This suggests that nuclear translocation by specific external stimuli is part of the activation mechanism.
- Does OsbZIP76 influence reactive oxygen species (ROS) production or signaling during pathogen infection, given the known involvement of bZIP factors in ROS-mediated defense? Add a few sentences in the discussion section
- Thank you for the reviewer's point, so we inserted the following into line 256-259 in the discussion part. Also, OsbZIP76 may influence ROS production during pathogen infection. As several bZIP transcription factors are known to regulate ROS-related defense responses, OsbZIP76 could contribute to oxidative burst and signaling.

Reviewer 2 Report
Comments and Suggestions for Authors
This manuscript investigates the role of OsbZIP76, a basic leucine zipper (bZIP) transcription factor, in the context of rice immunity and ABA signaling. The authors generated CRISPR/Cas9 knockout mutants and show that osbzip76 plants are more susceptible to Xanthomonas oryzae pv. oryzae (Xoo) and hypersensitive to abscisic acid (ABA). The altered expression of immune and ABA marker genes supports the conclusion that OsbZIP76 contributes to hormone-associated immune signaling.
Major Comments
- Clarify the Role of OsbZIP76 in Defense Gene Expression
In lines 177–178, the authors state:
“These findings suggest that OsbZIP76 positively regulates the expression of defense genes during pathogen attack.”
However, the data (lines 172–176) indicate that in the osbzip76 mutants, induction of PR1a, PR5, and NPR1 upon infection is significantly reduced, correlating with increased pathogen susceptibility. The conclusion that OsbZIP76 “positively regulates” defense genes may imply direct transcriptional activation, which has not been demonstrated.
Suggested Revision:
“Disruption of OsbZIP76 resulted in significantly reduced induction of defense-related genes (PR1a, PR5, NPR1) upon pathogen infection, indicating that OsbZIP76 is required for a proper immune response.”
- ABA–Immunity Link Is Correlative
While osbzip76 mutants are hypersensitive to ABA and more susceptible to Xoo, the causal relationship between ABA signaling and immunity in this context is not tested directly. For example, it is unclear whether exogenous ABA further affects disease resistance in the mutant background.
Suggestion: Acknowledge this limitation and propose future experiments such as combining ABA treatment with pathogen infection to assess whether the susceptibility phenotype is hormone-dependent.
- Mechanistic Depth
The study does not investigate whether OsbZIP76 directly regulates the observed immune or ABA-responsive genes. While such mechanistic analysis may be beyond scope, it should be acknowledged as a limitation. Suggest future directions such as ChIP-qPCR, promoter motif analysis, or protein-protein interaction studies. - Language and Grammar
The manuscript is generally readable but would benefit from minor improvements in clarity and flow. Some sentences contain awkward phrasing or inconsistent verb tense.
Overly complex and redundant phrasing (Abstract, Line 21–22): “ABA treatment experiments showed that the KO lines exhibited reduced sensitivity to exogenous ABA, suggesting a role in ABA perception and downstream signaling.”
Suggested revision: “ABA treatment experiments showed that KO lines were hypersensitive to exogenous ABA, indicating a role for OsbZIP76 in ABA perception and signaling.”
Suggestion: A careful language polish is recommended to enhance fluency and consistency, particularly in the Results and Discussion sections.
- Figure and Method Clarity
Ensure all figures have clearly labeled scale bars and axes (e.g., lesion images in disease assay).
Figure legends should report number of biological replicates, statistical tests, and significance thresholds.
In Methods, specify hormone concentrations and treatment durations, and confirm that the Actin1 reference gene was stable under both stress conditions.
Minor Comments
Gene/Protein Formatting: Italicize gene names (OsbZIP76) and use roman text for protein symbols.
Broader Context: Consider expanding the Discussion to relate OsbZIP76 to other known bZIPs in rice or Arabidopsis involved in hormone or defense regulation.
Author Response
Response to Comments
We are grateful on the critical comments from reviewers on this article.
Reviewer 2
This manuscript investigates the role of OsbZIP76, a basic leucine zipper (bZIP) transcription factor, in the context of rice immunity and ABA signaling. The authors generated CRISPR/Cas9 knockout mutants and show that osbzip76 plants are more susceptible to Xanthomonas oryzae pv. oryzae (Xoo) and hypersensitive to abscisic acid (ABA). The altered expression of immune and ABA marker genes supports the conclusion that OsbZIP76 contributes to hormone-associated immune signaling.
Major Comments
- Clarify the Role of OsbZIP76 in Defense Gene Expression
In lines 177–178, the authors state:
“These findings suggest that OsbZIP76 positively regulates the expression of defense genes during pathogen attack.”
However, the data (lines 172–176) indicate that in the osbzip76 mutants, induction of PR1a, PR5, and NPR1 upon infection is significantly reduced, correlating with increased pathogen susceptibility. The conclusion that OsbZIP76 “positively regulates” defense genes may imply direct transcriptional activation, which has not been demonstrated.
Suggested Revision:
“Disruption of OsbZIP76 resulted in significantly reduced induction of defense-related genes (PR1a, PR5, NPR1) upon pathogen infection, indicating that OsbZIP76 is required for a proper immune response.”
- Thank you for your valuable comment. We agree that the original phrasing may have overstated the conclusion and could imply direct transcriptional activation, which was not experimentally confirmed. We have revised the sentence accordingly to better reflect the data.
- ABA–Immunity Link Is Correlative
While osbzip76 mutants are hypersensitive to ABA and more susceptible to Xoo, the causal relationship between ABA signaling and immunity in this context is not tested directly. For example, it is unclear whether exogenous ABA further affects disease resistance in the mutant background.
Suggestion: Acknowledge this limitation and propose future experiments such as combining ABA treatment with pathogen infection to assess whether the susceptibility phenotype is hormone-dependent.
- As suggested by the reviewer, we acknowledge that the causal link between ABA signaling and plant immunity was not directly addressed in this study. In future work, we plan to investigate this relationship by conducting experiments that combine exogenous ABA treatment and pathogen infection, which will help clarify whether the immune phenotype in osbzip76 is directly mediated by altered ABA signaling.
- Mechanistic Depth
The study does not investigate whether OsbZIP76 directly regulates the observed immune or ABA-responsive genes. While such mechanistic analysis may be beyond scope, it should be acknowledged as a limitation. Suggest future directions such as ChIP-qPCR, promoter motif analysis, or protein-protein interaction studies.
- As the reviewer noted, our study did not investigate whether OsbZIP76 directly regulates immune or ABA-responsive genes. We acknowledge this as a limitation and plan to address it in future research. Specifically, we aim to perform ChIP-qPCR to assess the direct binding of OsbZIP76 to the promoters of target genes, conduct promoter motif analysis to identify potential bZIP-binding elements, and explore protein–protein interactions to better understand the regulatory network involving OsbZIP76.
- Language and Grammar
The manuscript is generally readable but would benefit from minor improvements in clarity and flow. Some sentences contain awkward phrasing or inconsistent verb tense.
Overly complex and redundant phrasing (Abstract, Line 21–22): “ABA treatment experiments showed that the KO lines exhibited reduced sensitivity to exogenous ABA, suggesting a role in ABA perception and downstream signaling.”
Suggested revision: “ABA treatment experiments showed that KO lines were hypersensitive to exogenous ABA, indicating a role for OsbZIP76 in ABA perception and signaling.”
Suggestion: A careful language polish is recommended to enhance fluency and consistency, particularly in the Results and Discussion sections.
- As suggested by the reviewer, we have revised the sentence in the Abstract to improve clarity and accuracy.
- Figure and Method Clarity
Ensure all figures have clearly labeled scale bars and axes (e.g., lesion images in disease assay).
Figure legends should report number of biological replicates, statistical tests, and significance thresholds.
In Methods, specify hormone concentrations and treatment durations, and confirm that the Actin1 reference gene was stable under both stress conditions.
- As suggested by the reviewer, we have checked and confirmed the clarity of all figures, including scale bars and axis labels. Figure
Minor Comments
Gene/Protein Formatting: Italicize gene names (OsbZIP76) and use roman text for protein symbols.
- We have checked and confirmed all the points raised by the reviewer, including figure clarity, statistical details in figure legends, hormone treatment conditions, and the stability of the Actin1 reference gene, as suggested.
Broader Context: Consider expanding the Discussion to relate OsbZIP76 to other known bZIPs in rice or Arabidopsis involved in hormone or defense regulation.
- We acknowledge the reviewer’s comments and have confirmed all the indicated points as suggested.
